# Repeated Measurements Are Necessary for Evaluating Accurate Diurnal Rhythm Using a Self-Intraocular Pressure Measurement Device

**DOI:** 10.3390/jcm12072460

**Published:** 2023-03-23

**Authors:** Yumi Shigemoto, Yuka Hasebe, Kazuyoshi Kitamura, Yoshiko Fukuda, Masako Sakamoto, Mio Matsubara, Shinya Minaguchi, Kenji Kashiwagi

**Affiliations:** Department of Ophthalmology, Faculty of Medicine, University of Yamanashi, Yamanashi 409-3898, Japan

**Keywords:** intraocular pressure, glaucoma, diurnal rhythm, self-measurement, reproducibility

## Abstract

Purpose: To investigate how many tests need to be performed to adequately assess intraocular pressure (IOP) diurnal change using a self-measuring rebound tonometer among glaucoma patients. Subjects and Methods: Adult patients with primary open-angle glaucoma were included. IOP was measured in the morning (6 AM to 9 AM), afternoon (12 PM to 3 PM), and at night (6 PM to 9 PM) for seven consecutive days. Twenty-four (7 males and 17 females, mean age 59.5 ± 11.0 years) patients who successfully measured IOP at least three times per day during the correct time periods for four days were subjected to analysis. Results: The IOP rhythm was significantly greater on the first day of measurement (6.6 ± 3.6 mmHg) than that averaged during subsequent days (4.4 ± 2.2 mmHg). The time of the highest and lowest IOP measurements on the first day of IOP measurement and during the entire measurement period coincided in 72.9% and 64.6% of cases, respectively. The concordance rate of the highest IOP time between the whole measurement period and each measurement day was less than 60%. Conclusion: The diurnal IOP rhythm measured by the patients themselves was not consistent, and multiple days of measurements may be necessary to correctly assess diurnal IOP rhythm.

## 1. Introduction

Only intraocular pressure (IOP) reduction proved to be an effective method for treating glaucoma. It is well known that IOP follows the diurnal rhythm [1]. An IOP measurement is usually performed during the daytime, and IOP elevation during nonbusiness hours is sometimes missed, which influences the onset or deterioration of glaucomatous optic neuropathy [2,3]. Therefore, it is important to accurately monitor IOP throughout the day, but this could be a burden for both patients and medical staff. Recent studies have shown that multiple days of measurements are necessary to correctly assess the pattern of diurnal rhythm in IOP [4,5,6,7].

In recent years, various tonometers have been developed. A self-measuring rebound tonometer (iCare Home^®^) enables patients for measuring IOP and could be useful to measure IOP much more frequently. Indeed, some previous papers reported that iCare Home^®^ could be useful to better understand the status of IOP [8,9,10,11,12]. However, some previous papers pointed out concerns under clinical usage [10,11,13,14,15]. Some patients could not measure their IOP correctly due to difficult handling or other reasons. Dabasia et al. reported that glaucoma patients were less likely than healthy subjects to have HOME measurements available [16].

In this study, we examined the consistency of IOP measurements using iCare Home^®^ and factors associated with this measurement and evaluated its usefulness in glaucoma care.

## 2. Patients and Methods

This study was approved by the University of Yamanashi Ethical Review Board. Since this study was a retrospective study, the ethics committee determined that individual consent was unnecessary. This study was conducted in accordance with the tenets of the Declaration of Helsinki. Self-measurements were performed routinely if ophthalmologists recognized they had to undertake this for clinical practice. The enrolled patients were subjects with primary open-angle glaucoma (POAG) who received outpatient glaucoma treatment at Yamanashi University Hospital from June 2016 to June 2017 and participated in the investigation of diurnal IOP variability by using iCare Home^®^. Patients with ophthalmic diseases other than mild cataracts, those who underwent intraocular surgery within 6 months, and those who failed to measure IOP at least three times a day for at least three days during the measurement period were excluded. Patients whose anti-glaucoma eye drops had been changed within one month were also excluded. Medical staff repeatedly instructed enrolled patients on how to handle iCare Home^®^ until patients could measure a reliable IOP. Patients continued to use their daily anti-glaucoma eye drops during the measurement period.

### 2.1. IOP Measurement

The schedule for IOP measurement was every 3 h from 6:00 to 24:00 (6, 9, 12, 15, 18, 21, 24:00) for seven days from Monday to Sunday. Patients were also instructed to record IOP-measurement time, wake-up time, sleep onset time, and when they used their anti-glaucoma eye drops. Patients measured their IOP in the sitting position. IOP measurements were performed according to the manufacturer’s instructions. Patients measured their IOP by continually pressing the button on the device using a single mode until they heard a long beep. This process was repeated until six measurements that were judged to be appropriate were obtained. If a single measurement was successfully finished, the “DONE” light illuminated the back panel of iCare Home^®^. After one week of measurements, the instruments were retrieved and analyzed by an ophthalmologist. The measurements were transferred to a personal computer installed with the iCare LINK software program (iCare, Finland Oy, Vantaa, Finland). Patients were requested to report the measured times, daily activities, and anti-glaucoma eye-drop usage times to confirm the data.

### 2.2. Classification of Diurnal IOP Rhythm

IOP values from 6 AM to 9 AM were defined as morning values, 12 PM to 6 PM as afternoon values, and 9 PM to 12 PM as nighttime values. The pattern of diurnal rhythm was classified into four types according to the time of day when the highest IOP was observed: morning elevation, afternoon elevation, nighttime elevation, and flat type. The flat type was defined as a maximum difference in IOP on the same day that was the same or less than 2 mmHg. In the current study, the magnitude of IOP fluctuation was not taken into account when differentiating the diurnal IOP rhythm pattern. Patients who could measure their IOP at least once during each time period for at least four days were subject to a reproducibility study of diurnal IOP rhythm.

### 2.3. Main Outcome Measures

The following points were examined: (1) the distribution of diurnal IOP rhythm during the measurement period; (2) the reproducibility of diurnal IOP rhythm; (3) the percentage of agreement between diurnal IOP rhythm; and (4) the analysis of IOP variability patterns and factors related to the reproducibility of diurnal IOP rhythm.

### 2.4. Statistical Analysis

Statistical analysis was performed on JMP (SAS Institute Inc., Cary, NC, USA). The Mann-Whitney *U* test, the Tukey-Kramer multiple comparison tests, multivariate regression analysis, or contingency table analysis was employed for the statistical analysis. *p* values less than 0.05 were considered significant. The values are presented as the means ± standard deviations.

## 3. Results

There were 33 subjects (11 men and 22 women) who met the inclusion criteria. The average age was 59.5 ± 11.0 years. Among these subjects, some patients were unable to obtain reliable IOP values at some measurement times. Therefore, only reliable IOP values were included in the study. Nine patients had primary open-angle glaucoma, and 24 patients had normal tension glaucoma. Their demographics are depicted in Table 1.

### 3.1. Diurnal IOP during the Investigated Period

The mean IOP values during the entire experimental period were 10.9 ± 3.8 mmHg in the right eye and 11.3 ± 4.2 mmHg in the left eye. There was no significant difference in IOP between the right eye and the left eye. Figure 1 shows the mean IOP at each time point. IOP was the highest at 6 AM and lowest at 12 PM, and the most significant difference in the Tukey–Kramer multiple comparison tests was between 6 AM and 12 PM in the right eye (*p* = 0.027).

### 3.2. Magnitude of Diurnal IOP Change

The mean magnitude of IOP change was 5.6 ± 3.68 mmHg in the right eye and 6.1 ± 4.09 mmHg in the left eye. The magnitude ranged from 0 to 16 mmHg. Figure 2 shows the distribution of the mean diurnal IOP change during the measurement period. The most frequent range of the mean diurnal IOP change was between 2 mmHg and 4 mmHg in approximately half of the patients. There was no significant difference between the right and left eyes.

### 3.3. Comparison of Patterns of Diurnal IOP Rhythm

We investigated changes in the pattern of diurnal IOP rhythm during the experimental period. The most frequent rhythm pattern was the morning elevation pattern in more than 40% of both eyes, followed by the afternoon elevation pattern and the nighttime elevation pattern in both eyes (Figure 3a,b). However, the pattern of diurnal IOP rhythm varied from day to day.

### 3.4. The Consistency of Diurnal IOP Rhythm and Its Associated Factors

We examined consistency, defining consistency as the same patterns of diurnal IOP rhythm on the first day and on the second and subsequent days. The consistency was approximately 50–60% in both eyes and tended to decline on day 7. There was no significant difference in diurnal IOP rhythm among the measured days in either eye. (Figure 4a,b). IOP values measured in the outpatient setting showed a positive relationship with the consistency of diurnal IOP rhythm, while the mean deviation (MD) from static visual field testing (Humphrey visual field meter program 24-2, 10-2, Carl Zeiss Meditec, Tokyo, Japan) showed a negative relationship with the consistency of diurnal IOP rhythm, although these relationships were not significant (*p*-value > 0.05). There were no significant associations between diurnal IOP rhythm and age, sex, best-corrected visual acuity, MD value, number of anti-glaucoma eye drops, type of glaucoma, or ambulatory IOP.

### 3.5. Magnitude of Fluctuation of Diurnal IOP Changes on Each Measurement Day

To examine the magnitude of the diurnal IOP rhythm in the measurement period, the standard deviation (SD) of the diurnal IOP rhythm on each measurement day was examined. The SD values were greater on the first two measured days and then gradually decreased during the experimental period. The SD values from day 3 through to day 6 were significantly smaller than those at day 1.

## 4. Discussion

The measurement of diurnal rhythm is important for understanding IOP status, but various concerns have been pointed out [1]. Diurnal IOP patterns obtained with rebound self-tonometry on two consecutive days showed inconsistency in between 47 and 63% of patients [6,17].

In the current study, IOP variability was assessed for a period of one week to more accurately examine the diurnal IOP change. The repeatability of the variability was moderate, and the agreement with the 7-day average variability pattern was moderate. This result is consistent with previous reports. A single diurnal IOP change may not be sufficient to accurately assess the rhythm pattern.

The reason for the variability in diurnal IOP was apparently due to measurement methods and other factors. As a result of investigating the magnitude of fluctuation in IOP, as indicated in Figure 5, the variability gradually decreased after the start of the measurement and was relatively stable after the third day of measurement. This indicates that the measured values stabilized after learning the measurement method. In this measurement, there were patients who had difficulty measuring their IOP despite thorough training on how to use the device. There are several possible reasons. A potential source of error observed by our research staff was a tendency for patients to decenter the probe onto the inferior cornea during measurements. It is not known how this may have affected unsupervised measurements performed at home. Decentered iCare measurements have been shown to underestimate pressure when compared to measurements correctly taken at the central cornea; in particular, measurements taken at the inferior cornea showed the poorest correlation with IOP measured with applanation tonometry [18,19]. Datasia et al. reported that glaucoma patients were less likely than healthy subjects to have home measurement devices available [16]. Generally, the complete success of a single measurement by patients with glaucoma or glaucoma suspects was estimated to be approximately 70%–75% [9,16]. Therefore, if glaucoma patients attempt diurnal IOP measurements, obtaining complete data will be more difficult. In this study, we investigated the factors that affect the diurnal IOP rhythm but did not find any significantly related factors. However, MD and IOP values showed a tendency to correlate. We speculate that older patients have less mobility; severe visual field defects, especially central visual field disturbance, may affect the accuracy of IOP measurement, although this hypothesis could not be confirmed in this study. Incomplete eyedrop instillation may influence the results, although we confirmed that patients instilled eyedrops as indicated by the patient’s report. These results indicate that further improvements to the iCare Home^®^ instrument design may be necessary to reduce the apparent measurement variability.

By contrast, it is possible that the diurnal IOP change is truly a day-to-day occurrence. Environmental factors such as diet may influence IOP. The number and duration of daily patient meals were noted in the questionnaire, but no detailed information on diet content was collected. Psychological stress may also affect diurnal rhythm because some stress hormones, including cortisol and the sympathetic nervous system, have been linked to the regulation of IOP.

In this study, we clarified that IOP was highest at 6:00 AM and lowest at 12:00 PM and that there was considerable individual rhythm in the pattern of IOP fluctuation. Most individuals had the highest IOP in the morning, which indicates that we may have missed daily maximum IOP measurements by conducting only casual IOP measurements. The magnitudes of diurnal IOP changes were 5.6 ± 3.68 mmHg in the right eye and 6.1 ± 4.09 mmHg in the left eye. Some patients showed a daily IOP change of more than 10 mmHg, and their daily IOP was relatively good. Although the evaluation of diurnal IOP measurement using iCare Home^®^ may be worthful even among patients showing relatively good IOP control, the current results indicate that repeated trials may be necessary to evaluate a diurnal IOP rhythm correctly.

The limitations of this study include the relatively small sample size and the use of only glaucoma subjects. The subjects continued using glaucoma eye drops, and they were asked to keep a diary of their eye drops and living conditions but were not restricted in their daily activities. Therefore, it is not clear whether the present results can be applied to eyes with nonglaucoma or those not treated with IOP-lowering eye drops. It is necessary to improve on these points in the future to study more accurate diurnal changes in IOP. The IOP values measured by GAT and iCare HOME are not interchangeable, as they use different methodologies to measure IOP. It is important not to directly compare the IOP values obtained from these two devices. Moreover, since there are no clear criteria for assessing the appropriateness of the number of IOP diurnal rhythm measurements, it is unclear from these results alone how many days the measurements need to be repeated to accurately assess IOP diurnal rhythm.

The measurement variance was large in the beginning but stabilized with repeated measurements. This may lead to the recommendation of taking multiple measurements to obtain a more accurate evaluation of the diurnal rhythm with iCare Home^®^. Clinicians can support the diagnosis and management of patients at risk of glaucoma by determining their peak IOP, as well as the extent and pattern of fluctuations over at least three consecutive days. For most patients, self-tonometry could be performed, although design modifications may improve the accuracy and ease of correct self-alignment, enabling more patients to use this technology. In the future, longitudinal studies utilizing rebound self-tonometry could help uncover the association between diurnal IOP fluctuations and glaucoma conversion and progression.

## Figures and Tables

**Figure 1 jcm-12-02460-f001:**
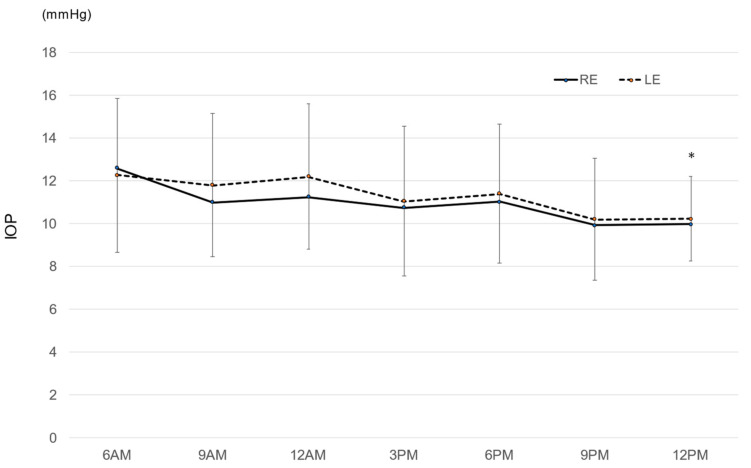
Mean diurnal intraocular pressure. * *p* = 0.027 6 AM vs. 12 PM, the Tukey–Kramer multiple comparison test. RE: right eye, LE: left eye, bar = standard deviation.

**Figure 2 jcm-12-02460-f002:**
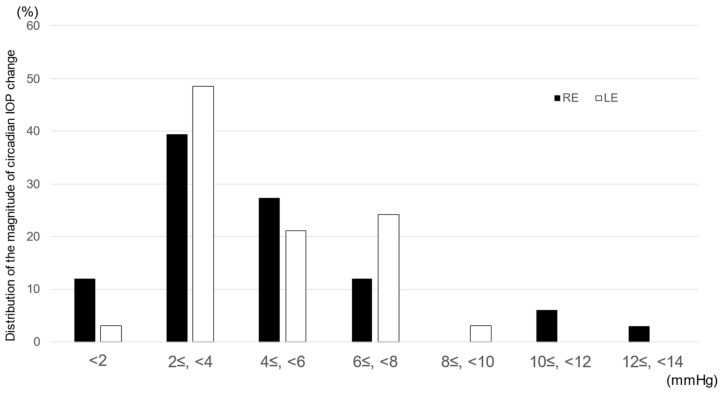
Distribution of mean diurnal intraocular pressure change during the measurement period. RE: right eye, LE: left eye.

**Figure 3 jcm-12-02460-f003:**
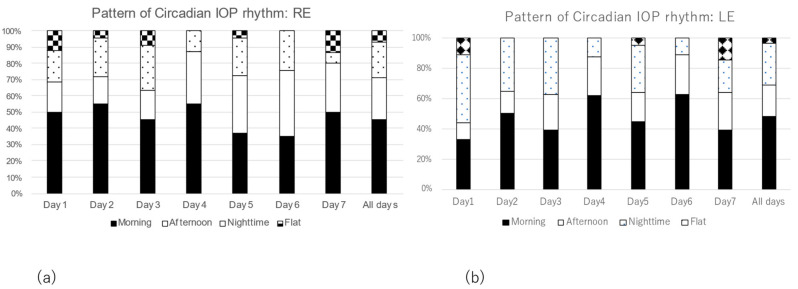
Changes in the pattern of diurnal IOP rhythm during the measurement period. (**a**) Right eye, (**b**) Left eye.

**Figure 4 jcm-12-02460-f004:**
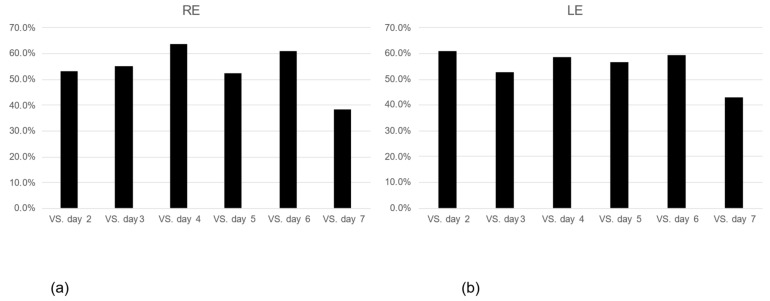
Consistency of the pattern of diurnal IOP change. (**a**) Right eye, (**b**) Left eye, *p* > 0.20, the Tukey–Kramer multiple comparison test.

**Figure 5 jcm-12-02460-f005:**
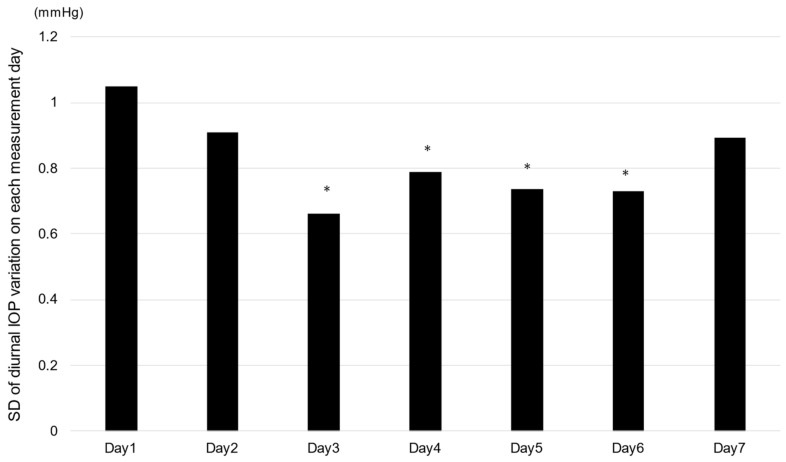
Fluctuation in the magnitude of diurnal IOP change. SD: standard deviation, * *p* < 0.05 vs. day 1, the Tukey–Kramer multiple comparison test.

**Table 1 jcm-12-02460-t001:** Demographics of enrolled patients.

	RE	LE
logMAR	0.072	0.11
	(range: −0.176~1.301)	(range: −0.176~1.222)
HFA-MD (24-2·30-2) (dB)	−11.2	−10.8
HFA-MD (10-2) (dB)	−12.4	−12.3
# of anti-glaucoma eyedrops	2.5	2.4
Mean ambulatory IOP (GAT) (mmHg)	12.4	12.4
	(range: 9.3~17.5)	(range: 8.0~17.3)
CCT (um)	511 ± 186.4	515.3 ± 188.9

RE: right eye, LE: left eye, LogMAR: logarithm of the minimum angle of resolution, HFA-MD: Humphrey Field Analyzer-mean deviation, GAT: Goldmann Applanation Tonometer, CCT: central corneal thickness.

## Data Availability

The datasets generated and/or analyzed during the current study are available from the corresponding author on reasonable request.

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
