# Peer review of "Repeated Measurements Are Necessary for Evaluating Accurate Diurnal Rhythm Using a Self-Intraocular Pressure Measurement Device"

_jcm, 2023, doi:10.3390/jcm12072460_

Round 1
Reviewer 1 Report (Previous Reviewer 2)
The authors responded well to the reviewer's points.
Lines136-137. Please add P values in the main texts or in Figure 4 to show how unsignificant the difference was in the diurnal IOP rhythm.
Author Response
Thank you for your comment. I added P value in figure legend.
Reviewer 2 Report (Previous Reviewer 1)
Thank you all comments were adequately revised.
Author Response
Thank you for your evaluation. I deeply appreciate your kindly review.
This manuscript is a resubmission of an earlier submission. The following is a list of the peer review reports and author responses from that submission.
Round 1
Reviewer 1 Report
There is a Typo in the title “How many repetitions ARE necessary…”
Regarding the study design: The authors stated that the analyses were done retrospectively. I am wondering if self-measurements are performed routinely or if the patients participated in a different earlier trial?
Did you perform Goldmann tonometry as well? In the title you mention “accurate circadian rhythm”. What do this refer to? Accurate as compared to the gold standard measurement of intra ocular pressure? Please comment
One aspect is left out completely. How many measurements were actually performed during each time point? What was the patients` adherence to the measurement protocol and how many measurements were classified as successful? You mentioned correctly that only 70-75% are successful in previous studies. Please add the information, how many measurements were analysed, to every analysis (especially within the figures)
Axis labelling is missing in every figure.
In the results section the authors present different aspects. A few of them are not clear to me:
Please add relevant p – values
Magnitude of circadian IOP change: do you mean the maximal difference during each day? During the whole week? Please clarify
Comparison of different patterns of IOP rhythm describes when the IOP increases? Or when IOP was highest? Criteria in that regard are missing. Please add this information.
In figure 2 the authors show that approximately 10% of right eyes have less variation than 2mmHg, who were classified as “flat”, correct? However, in figure 3 the rate of “Flat” is below 10%. Please clarify.
How do you assess consistency of circadian IOP? The percent values are not clear. What do they describe please comment.
Why do you not present SD straight away? Please comment
L149 in the text the authors mention that day 2 was significant (no p values are presented!). In figure 5 Day 2 is not marked with an asterisk.
Most part of your results are not reflected in the discussion. Please comment.
L190 “Patients were fed at each of the visits” what do you mean by that?
L200 The presence of higher variability in your data cannot be linked directly to the progression, as this was not part of the study. Please rephrase
Author Response
Dear reviewer 1
Thank you for your kindly comments. I revised some points according to your comments as below. Major revised points are highlighted in yellow background.
Reviewer’s comment
There is a Typo in the title “How many repetitions ARE necessary…”
Author’s reply
I fixed this point.
Reviewer’s comment
Regarding the study design: The authors stated that the analyses were done retrospectively. I am wondering if self-measurements are performed routinely or if the patients participated in a different earlier trial?
Author’s reply
I added the following description.
Self-measurements are performed routinely if ophthalmologists recognize to do this for clinical practice.
Reviewer’s comment
Did you perform Goldmann tonometry as well? In the title you mention “accurate circadian rhythm”. What do this refer to? Accurate as compared to the gold standard measurement of intra ocular pressure? Please comment
Author’s reply:
We routinely measure IOP using Goldmann tonometry in the outpatient clinic. However, we cannot simply compare IOP values measured by Goldmann tonometry and iCare Home, since IOP measurements were performed at different hours between two methods. The main subject of this paper is to clarify how many reputation is required to obtain accurate circadian IOP change if iCare Home, since previous studies reported that a single measurement of iCare Home could not provide accurate circadian IOP rhythm.
Reviewer’s comment
One aspect is left out completely. How many measurements were actually performed during each time point? What was the patients` adherence to the measurement protocol and how many measurements were classified as successful? You mentioned correctly that only 70-75% are successful in previous studies. Please add the information, how many measurements were analysed, to every analysis (especially within the figures)
Author’s reply
As we write in the previous version.
Measurement process was repeated until six measurements that were judged to be appropriate were obtained. If a single measurement was successfully finished, the “DONE” light illuminated the back panel of iCare Home®.
Since the real number of each measurement is not same among patients or measurements, it is difficult to show how many measurement patients require to complete single measurement.
The meaning of only 70-75% success in prior studies is the percentage of cases in which the IOP measurements required to study diurnal variation could not be completed, not the success rate of each IOP measurement.
Reviewer’s comment
Axis labelling is missing in every figure.
Author’s reply
I added axis labeling in all figures.
Reviewer’s comment
In the results section the authors present different aspects. A few of them are not clear to me:
Author’s reply
I revised some points for better understanding.
Reviewer’s comment
Please add relevant p – values
Author’s reply
I added some additional information.
Reviewer’s comment
Magnitude of circadian IOP change: do you mean the maximal difference during each day? During the whole week? Please clarify
Author’s reply
I revised some points for better understanding.
Reviewer’s comment
Comparison of different patterns of IOP rhythm describes when the IOP increases? Or when IOP was highest? Criteria in that regard are missing. Please add this information.
Author’s reply
I added the following description.
In the current study, the magnitude of IOP fluctuation was not taken into account when differentiating the diurnal IOP variation pattern.
Reviewer’s comment
In figure 2 the authors show that approximately 10% of right eyes have less variation than 2mmHg, who were classified as “flat”, correct? However, in figure 3 the rate of “Flat” is below 10%. Please clarify.
Author’s reply
For the evaluation of reproducibility of diurnal variation patterns depicted in Figure3, patients who could measure their IOP at least once during each time period for at least four days are chosen, while in the case of evaluation of IOP variation range depicted in Figure 2, all patients are chosen, which results in discrepancies between the results in Figures 2 and 3.
Reviewer’s comment
How do you assess consistency of circadian IOP? The percent values are not clear. What do they describe please comment.
Author’s reply
Unfortunately, there are no clear criteria for evaluating reproducibility. However, the current results such as relatively low reproducibility shown in Figure 3, the fluctuation of magnitude of circadian IOP change shown in Figure 5, and not so high consistency of circadian IOP rhythm pattern shown in Figure 4 suggest that repeated measurements are necessary for proper evaluation of circadian IOP change at least using iCare Home as described in the discussion.
I added the following description.
Moreover, since there are no clear criteria for assessing the appropriateness of the number of IOP diurnal variation measurements, it is unclear from these results alone how many days the measurements need to be repeated to accurately assess IOP diurnal variation.
Reviewer’s comment
Why do you not present SD straight away? Please comment
Author’s reply
Figure 5 shows the standard deviation (SD) of diurnal IOP variation on each measurement day to investigate the magnitude of the circadian IOP variation in the measurement period. I am so sorry that our previous paper failed to explain this analysis precisely. Actually, Y axis in Figure 5 presents standard deviation of diurnal IOP on each measurement day.
I revised as followed at page 8, lines 120.
Reviewer’s comment
L149 in the text the authors mention that day 2 was significant (no p values are presented!). In figure 5 Day 2 is not marked with an asterisk.
Author’s reply.
I am sorry that I simply miswrite this part. I revised this as followed.
The SD values at day 3 through day 6 were significantly smaller than those at day 1.
Reviewer’s comment
Most part of your results are not reflected in the discussion. Please comment.
Author’s reply.
I revised our discussion part as indicated.
Reviewer’s comment
L190 “Patients were fed at each of the visits” what do you mean by that?
Author’s reply
I am sorry about this. I revised as followed.
The number and duration of daily patient meals were noted in the questionnaire, but no detailed information on diet content was collected.
Reviewer’s comment
L200 The presence of higher variability in your data cannot be linked directly to the progression, as this was not part of the study. Please rephrase
Author’s reply
I revised as followed.
Therefore, evaluation of circadian IOP measurement using iCate Home® may be worthful even among patients showing relatively good IOP control.
Reviewer 2 Report
This study is about a very interesting issue, but has many flaws that need to be improved.
The study starts with questioning how many repetitions are necessary, for which the answer is not provided in the Conclusion. Authors only conclude that multiple days of measurements are necessary, and did not provide guidelines as to how many days would be necessary for patients' learning period, when could reliable measurements be obtained after this learning period, and then how many days of measurements should be considered to obtain reliable values.
Measurements on first day seem to have largest variability and inconsistency between measurements. However, it seems that the study compares the IOP measurements based on the first day values.
Specific points
1. Line 57: please clarify how "those who could not be reliably measured by iCare Home" was defined.
2. Lines 80-81: Specify accurately the criteria for the classification.
3. Line 133: Clarify what "similar" exactly means.
4. Lines 135-137: This sentence is not clear to this reviewer. Please provide more detailed explanation with supporting data.
Author Response
Reviewer 2
Thank you for your kindly comments. I revised some points according to your comments as below. Major revised points are highlighted in yellow background.
Reviewer’s comment
The study starts with questioning how many repetitions are necessary, for which the answer is not provided in the Conclusion. Authors only conclude that multiple days of measurements are necessary, and did not provide guidelines as to how many days would be necessary for patients' learning period, when could reliable measurements be obtained after this learning period, and then how many days of measurements should be considered to obtain reliable values.
Measurements on first day seem to have largest variability and inconsistency between measurements. However, it seems that the study compares the IOP measurements based on the first day values.
Author’s reply
Thank you for your comment. I agree with your comment. Unfortunately, there are no clear criteria for evaluating reproducibility. However, the current results such as relatively low reproducibility shown in Figure 3, the fluctuation of magnitude of circadian IOP change shown in Figure 5, and not so high consistency of circadian IOP rhythm pattern shown in Figure 4 suggest that repeated measurements are necessary for proper evaluation of circadian IOP change at least using iCare Home as described in the discussion.
I added the following description.
Moreover, since there are no clear criteria for assessing the appropriateness of the number of IOP diurnal variation measurements, it is unclear from these results alone how many days the measurements need to be repeated to accurately assess IOP diurnal variation.
Reviewer’s comment
- Line 57: please clarify how "those who could not be reliably measured by iCare Home" was defined.
Author’s reply
I revised as followed.
Patients who failed to measure IOP at least three times a day for at least three days during the measurement period were excluded.
Reviewer’s comment
- Lines 80-81: Specify accurately the criteria for the classification.
Author’s reply
I revised as followed.
The pattern of circadian rhythm was classified into four types: according to the time of day when the highest IOP was observed: morning elevation, afternoon elevation, nighttime elevation, and flat type.
Reviewer’s comment
- Line 133: Clarify what "similar" exactly means.
Author’s reply
I replaced similar by same.
Reviewer’s comment
- Lines 135-137: This sentence is not clear to this reviewer. Please provide more detailed explanation with supporting data.
Author’s reply
I revised this point as followed at page
IOP values measured in the outpatient setting showed a positive relationship with consistency of circadian IOP rhythm, while mean deviation (MD) from static visual field testing (Humphrey visual field meter program 24-2, 10-2, Kurzweiss Meditec, Tokyo, Japan) showed a negative relationship with consistency of circadian IOP rhythm, although these relationships were not significant.
Round 2
Reviewer 1 Report
Thank you very much for your comments and replies.
Structure of the new manuscript is shuffled. (Methods after discussion)
As part of my previous comment, I recommend adding the information how many patients completed each measurement successfully.
I the revised version of the manuscript the authors added the information that patients were included if the completed at least three measurements during the day on at least three days a week. This means that not every patient measured during every time point. Please ad this information.
You explained the difference between Figure 3 and Figure 2. The patients or measurements included into the respective figure is still not clear for the reader. Please ad the information to the figure legend and into the text.
Figure 5 unit of y – axis is missing (mmHG).
L182 again typo "iCate"
L182 Why do you think this new statement is supported by your findings? You could write that reproducibility might be better after the third day, but again you cannot link you finding to findings not investigated into the study.
Author Response
Dear reviewer 1
Thank you for your kindly comments. I revised some points according to your comments as below. Major revised points are highlighted in yellow background.
Reviewer’s comment
Structure of the new manuscript is shuffled. (Methods after discussion)
Author’s reply
I am sorry about this. I change the order as pointed out.
Reviewer’s comment
As part of my previous comment, I recommend adding the information how many patients completed each measurement successfully.
In the revised version of the manuscript the authors added the information that patients were included if the completed at least three measurements during the day on at least three days a week. This means that not every patient measured during every time point. Please ad this information.
Author’s reply
Thank you for your comments. Since some subjects failed to measure reliable IOP at some scheduled time points, we employed reliable IOP values for the analysis. I added the following sentences in the first paragraph of results.
“Among these subjects, some patients were unable to obtain reliable IOP values at some measurement times. Therefore, only reliable IOP values were included in the study.”
Reviewer’s comment
You explained the difference between Figure 3 and Figure 2. The patients or measurements included into the respective figure is still not clear for the reader. Please ad the information to the figure legend and into the text.
Author’s reply
I added the following description both line 140 and 151-152 in the text. I also change titles of figures 2 and 3.
Reviewer’s comment
Figure 5 unit of y – axis is missing (mmHG).
Author’s reply
I am sorry about this. I added unit of Y axis.
Reviewer’s comment
L182 again typo "iCate"
Author’s reply
I am sorry about this. I fixed it.
Reviewer’s comment
L182 Why do you think this new statement is supported by your findings? You could write that reproducibility might be better after the third day, but again you cannot link you finding to findings not investigated into the study.
Author’s reply
Thank you for your comment. I revise this portion as below.
Although evaluation of circadian IOP measurement using iCare Home® may be worthful even among patients showing relatively good IOP control, the current results indicate that repeated trials may necessary to evaluate a circadian IOP rhythm correctly.
Reviewer 2 Report
1. Please address my point below, which had not been resolved in the authors’ response.
“Measurements on first day seem to have largest variability and inconsistency between measurements. However, it seems that the study compares the IOP measurements based on the first day values.”
2. Lines 236-239 What was the definition of “flat type”? Does that mean IOP was the same for 24 hours of the day?
3. Lines 106-113. If the authors found the relationship significant or not significant, they should provide statistical values showing the relationships.
Author Response
Dear reviewer 2
Thank you for your kindly comments. I revised some points according to your comments as below. Major revised points are highlighted in yellow background.
Reviewer’s comment
Measurements on first day seem to have largest variability and inconsistency between measurements. However, it seems that the study compares the IOP measurements based on the first day values.”
Author’s reply
Thank you for your comment. As you comment, we compared consistency of circadian IOP changes during measurment period based on the first day result. However, other comparison analysis based on average values during measeurment period also showed a similar tendency that indicates circadian IOP change subject to relatively big fluctuation especially early trial days. I appreciate your understanding.
Reviewer’s comment
Lines 236-239 What was the definition of “flat type”? Does that mean IOP was the same for 24 hours of the day?
Author’s reply
Thank you for your comment. I changed the sentence as below for better understanding,
Flat type was defined that maximum difference in IOP in the same day was the same or less than 2 mmHg.
Reviewer’s comment
Lines 106-113. If the authors found the relationship significant or not significant, they should provide statistical values showing the relationships.
Author’s reply
I added (p value ,0.05)